# A 0.49–4.34 µW LC-SAR Hybrid ADC with a 10.85-Bit ENOB and 20 KS/s Bandwidth

Hai Tang [1], Weilin Xu [1,*], Haiou Li [2], Baolin Wei [2] and Xueming Wei [2]

1   Key Laboratory of Microelectronic Devices and Integrated Circuits, Education Department of Guangxi Zhuang Autonomous Region, Guilin University of Electronic Technology, Guilin 541004, China; 1800200730@mails.guet.edu.cn

2   Guangxi Key Laboratory of Precision Navigation Technology and Applications, Guilin University of Electronic Technology, Guilin 541004, China; lihaiou@guet.edu.cn (H.L.); blwei@guet.edu.cn (B.W.); scuweixue@guet.edu.cn (X.W.)

*   Correspondence: xwl@guet.edu.cn

**Abstract:** This paper presents a level-crossing successive-approximation-register (LC-SAR) hybrid analog-to-digital converter (ADC) that combines an LC ADC with an SAR ADC, which may be used for Internet of Things (IoT) random sparse event scenarios. The sampling frequency of a traditional LC ADC is usually proportional to the maximum instantaneous rate of change of the input signal; therefore, a higher input signal frequency inevitably leads to higher system power consumption. However, the proposed hybrid ADC uses the input level difference between the two moments before and after level-crossing detection, thereby ensuring a higher conversion precision and lower power consumption, even at higher input signal frequencies. Compared with traditional LC ADC or SAR ADC, the proposed hybrid ADC combines the ultralow-power advantage of LC ADC with the high-precision advantage of SAR ADC in converting IoT data with sparse characteristics such as ECG, EEG, and brain potential. The LC-SAR hybrid ADC is designed with a 0.18 µm CMOS process and consumes 4.34 µW at a 1.8 V supply voltage, achieving an SNDR of 67.41 dB and a bandwidth of 20 kHz. The spectrum analysis result was 10.85 ENOB when the input sinusoidal signal was 14.975 kHz. When inputted with an ECG signal, the system power consumption was as low as 0.49 µW. Furthermore, the proposed hybrid ADC obtained a good figure of merit, with FoMw and FoMs reaching 58.8 fJ/conv.steps and 164 dB, respectively. Compared to a conventional uniform sampling ADC, approximately 80% of the power savings and an 8x compression ratio can be achieved in physiological signal acquisition applications.

**Keywords:** level crossing; successive approximation register; hybrid; ADC; physiological signal

## 1. Introduction

According to research conducted by the World Health Organization (WHO), cardiovascular diseases are causing an increasing number of deaths [1]. Consequently, in mobile Internet of Things (IoT), portable medical equipment, such as wearable chronic disease monitors, electronic bracelets, and wristwatches, is an important application [2]. An essential module of the detection link, the analog-to-digital converter (ADC), converts an analog signal into a digital signal, enabling digital-signal-processing systems to process signals in the digital domain. In mobile application scenarios, the accuracy and speed of the ADCs in the system guarantee the portability and reliability of portable medical testing equipment, while ultralow power consumption ensures long endurance. High sparsity is a characteristic of physiological signals in the time domain. They can generate dramatic and informative changes at only one or a few points over long periods. The utilization of the conventional Nyquist uniform sampling method to acquire these signals leads to a significant amount of redundant sampling, thereby causing unnecessary energy consumption and low energy efficiency in the ADC. Level-crossing (LC) sampling is an event-driven

sampling technique that only samples a signal when its amplitude exceeds a specified threshold. This method is particularly well suited for the acquisition of physiological signals that are in the time domain and contain high sparsity [3]. For a segment of the ECG signal, Figure 1a,b illustrates the distinction between level-crossing sampling and uniform Nyquist sampling. The uniform sampling procedure (which applies equal time-interval sampling to an entire segment of the ECG signals) for this segment of signals is illustrated in Figure 1a. When applied to ECG signals characterized by high sparsity, this sampling technique generates redundant samples during each physiological signal cycle, thereby diminishing the energy efficiency of the ADC. Level-crossing sampling only samples when the signal undergoes a significant change with the QRS segment signal, as illustrated in Figure 1b. The level-crossing sampling method samples the signal only when it undergoes a significant change in intensity, as shown in Figure 1b. This resulted in a significantly greater number of samples in the QRS segment and a significantly reduced number of samples at other times, thereby enhancing the energy efficiency of the ADC. Therefore, it is essential to research LC ADCs utilizing the level-crossing sampling technique, which is more suitable for medical testing equipment applications requiring ADCs with minimal power consumption.

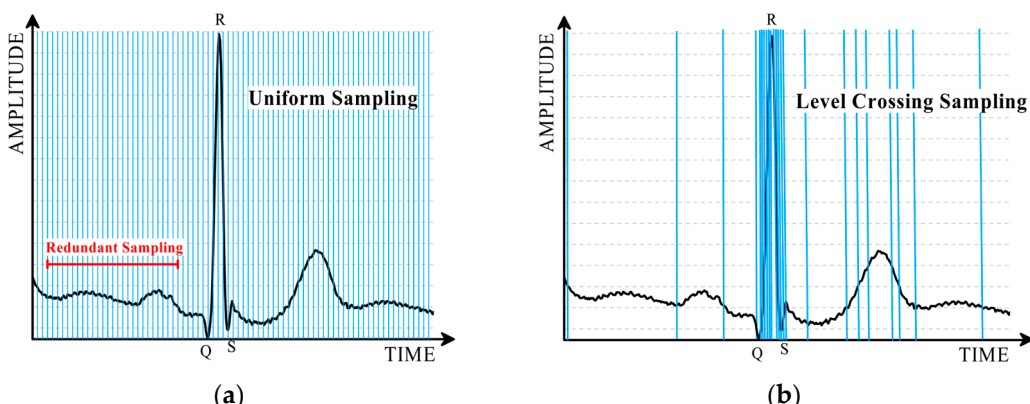

**Figure 1.** (**a**) Uniform sampling of ECG signals. (**b**) Level-crossing sampling of ECG signals.

The architecture of a conventional LC ADC is illustrated in Figure 2a,b. The fixed-window-type LC ADC [4] is shown in Figure 2a. Its circuit comprises an input charge-scaling circuit, comparator, and up/down counter. In contrast, a floating-window-type LC ADC [5] is depicted in Figure 2b and features a DAC, comparator, and up/down counter. In classic LC ADC designs, there is often a tradeoff between accuracy, power consumption, and speed.

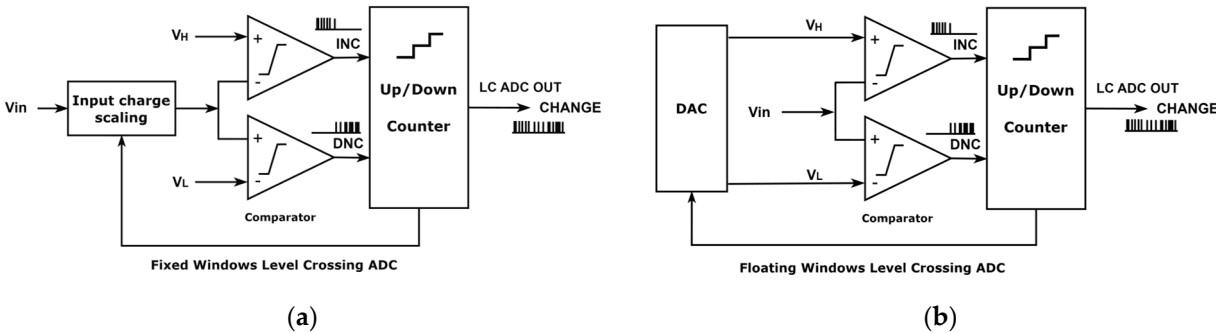

**Figure 2.** (**a**) Fixed-window LC ADC. (**b**) Floating-window LC ADC.

Although several nW-class LC ADCs have been presented with more than 8-bit accuracy [4,6,7], their bandwidth was limited to 1 kHz. This bandwidth is adequate for common ECG signals because their frequencies are typically below 500 Hz. However, they do not

meet the demands of physiological signals with higher frequencies, such as EEG and neural signals. The authors of Refs. [8–10] have developed several architectures with bandwidths up to 20 kHz and an accuracy greater than 8 bits; among these, Refs. [9,10] achieved an accuracy greater than 10 bits at the expense of a large power consumption. Therefore, for LC ADCs, tradeoffs must be made between the power consumption and speed. By employing an SAR ADC as an auxiliary ADC, the architecture suggested in Ref. [10] resolves the issue of distortion introduced by loop delays in conventional LC ADCs. It attains effective bit rates exceeding 10 bits and a bandwidth of 25 kHz, albeit at the expense of 5.7 µW power. The adaptive system clock was implemented in Ref. [9], wherein the system functions with diminished clock frequency when the input signal frequency is low. Despite attaining a bandwidth of 20 kHz, the system consumes 5.38 µW of power and requires an external FPGA to manage the clock and generate digital codes. Ref. [11] proposed a pure SAR architecture with a real-time QRS detector for ECG signal acquisition, where a dynamic tracking algorithm can track QRS segment signals while acquiring ECG signals. It obtains an ENOB of 10.72 bits, with only 1.88 µW of power consumption; however, its bandwidth was only 5 kHz. ADCs employing a purely SAR architecture fall short of achieving optimal performance when acquiring sparse signals such as those for ECG.

An event-driven quasi-level-crossing delta modulator was developed in Ref. [12] that has a bandwidth of up to 1.42 MHz and requires only 205 µW of power. However, the SNDR was only 53 dB. An ultralow-power ADC suitable for low-frequency IoT signal conversion still requires further improvement. Refs. [5,13] presented an LC-ADC with an asynchronous pipelined and event-driven architecture. It consumes only 41 nW of standby power owing to the wake-up function and demonstrates a higher energy economy of level cross-sampling in sparse events. Refs. [14,15] report on a wake-up circuit based on LC-ADC and pattern recognition. It can be utilized in wearable devices and consumes only 2.1 µW at a 2.6 kHz bandwidth. However, its ENOB was only 5.39 bits.

Ref. [16] presented a two-stage "5 + 5" C-ADC architecture that is more energy efficient than the LC-ADC while maintaining the same precision. The power consumption at bandwidths of 1 Hz–200 kHz is 160–426 µW. Ref. [17] proposed a fixed-window-type LC-ADC that consumes 30 µW at a bandwidth of 20 kHz. Their energy efficiency requires further improvements for wearable applications.

Ref. [18] presented an algorithm that can be used to detect the R-peak of the QRS segment during the measurement of the ECG signal that achieves very high accuracy and sensitivity when simulated with the MIT-BIH arrhythmia database. Refs. [19,20] described a logarithmic LC-ADC that differs from the conventional LC-ADC in that the DAC used for input charge scaling employs a logarithmic charge allocation method with lower INL and DNL errors. This work expands the idea of the research direction of LC-ADCs. Although its INL and DNL are lower than 0.17 LSB, it has only 3-bit precision and consumes 42.7 µW of power. Ref. [21] discussed a design that uses the level-crossing theory to control flash ADC sampling. The sampling signal regulates the sampling process of a 5-bit flash ADC, allowing the converter to effectively capture sparse signals. The auxiliary circuit is a floating-window-type LC ADC that consumes 506.9 µW. Actually, a single LC ADC must operate at a high frequency to accurately capture the instantaneous rate of change of the input signal as its frequency increases. If the LC ADC functions at a low frequency, its up/down counter may not accurately follow the input signal, which is a common issue in LC ADCs.

Although it is apparent that the LC ADC is better suited for time-domain sparse-signal acquisition, the current structure has several challenges. First, the converted CHANGE signal was used as a timing pulse, necessitating the regeneration of the digital code. Second, when the input signal frequency is increased, the LC ADC must operate at a very high sampling frequency, resulting in higher power consumption and a harder tradeoff between the bandwidth and power consumption.

In this study, an architecture combining an SAR ADC and an LC ADC is designed to solve the issue that the traditional LC ADC needs to work at a high sampling frequency to

meet the requirement of the change rate of the input signal when the input signal frequency is high. The architecture used in this study is similar to the architecture in Ref. [21], which utilized an LC ADC to generate sample signals for the SAR ADC. However, unlike Ref. [21], the proposed architecture relies on the SAR ADC output to help the LC ADC perform level-crossing detection. The designed LC ADC can function at a low sampling frequency without considering the rate of change requirement for the input signal. The SAR ADC architecture is characterized by high precision and low power consumption. The Nyquist ADC is unsuitable for sampling signals with a significant timing sparsity. Nevertheless, its limitations can be mitigated using an LC ADC. The proposed ADC is constructed using a 180 nm CMOS process, features an ENOB of 10.85 bits, and exhibits dynamic power consumption that varies between 0.49 μW when an ECG signal is input and 4.34 μW when a complete bandwidth of 20 kHz is input. The proposed hybrid architecture reaches an SNDR of 67.41 dB over a 20 kHz bandwidth and an FoM value of 164 dB. In addition, the proposed ADC generates a digital code output directly rather than a CHANGE clock pulse, enabling subsequent digital signal processing to occur directly. The remainder of this paper is organized as follows. The overall system architecture of the proposed level-crossing successive-approximation-register (LC-SAR) hybrid ADC chip is described in Section 2. Section 3 discusses in detail the circuit implementation of the LC-SAR ADC. The simulation results and analysis are presented in Section 4. Finally, Section 5 concludes the paper.

## 2. Proposed LC-SAR Hybrid ADC Architecture

### 2.1. Motivation

When the input signal frequency is high, conventional LC ADCs usually need to operate at a very high sampling frequency with high power consumption or maintain lower accuracy to meet the maximum instantaneous voltage change requirement of the signal. A 1.8 $V_{PP}$, 20 kHz sinusoidal signal is shown in Figure 3. The maximum instantaneous change in the signal voltage $\Delta V = 300$ mV at $\tau \approx 2.5$ us. Thus, the maximum instantaneous rate of change $\nu$ of this sinusoidal signal is obtained as follows

$$\nu = \frac{\Delta V}{\tau} = 120(\text{mV}/\mu\text{s}) \tag{1}$$

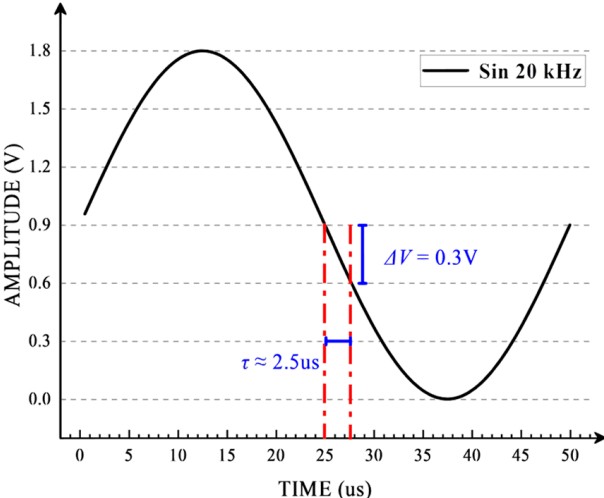

**Figure 3.** A 1.8 $V_{PP}$, 20 kHz sinusoidal signal and its maximum instantaneous voltage change.

Typically, for an LC ADC with N-bit hardware precision, the value of the signal voltage LSB that can be folded each time is

$$LSB = \frac{V_{ref}}{2^{N+1}} \tag{2}$$

When $V_{ref}$ = 1.8 V and N = 8, the LSB is approximately 3.5 mV. To meet the input signal change-rate requirement, the sampling frequency of the LC ADC ($FS_{LCADC}$) should satisfy the following expression

$$FS_{LCADC} \geq \frac{v}{LSB} \tag{3}$$

In other words, the $FS_{LCADC}$ must operate at a minimum of 34.3 MHz, which is an extremely high frequency. Furthermore, the charge scaling circuit in the LC ADC typically comprises capacitors, resulting in large dynamic power consumption. It is possible to decrease this frequency to save power using a larger LSB. However, this significantly reduces the accuracy of the LC ADC, which is unacceptable.

Thus, determining whether the LC ADC can continue to satisfy the rate change for the input signal requirement while operating at a reduced frequency is an issue that must be resolved. This study proposes an architecture that combines SAR and LC, as indicated by the above analysis. The conversion outcome of the SAR ADC with respect to the input signal of the preceding moment is currently used as the input of the LC ADC. The level-crossing detection function is then executed based on the voltage difference between the current input signal and input signal of the preceding moment. This enables the LC ADC to function at lower frequencies while fulfilling the input signal rate-of-change requirements. Furthermore, it renders the system output autonomous with respect to the LC ADC. The datatype that can be processed by the digital signal processing system is N-bit digital code, which is similar to the N-bit digital code produced by the conventional SAR ADC. However, the LC ADC outputs a segment of the timing pulse containing amplitude information. Therefore, data regeneration normally calls for the LC ADC output, which requires a signal representing the direction of change and step size of each step, in addition to the timing pulse output from the LC ADC. However, the architecture proposed in this study combines SAR and LC and obviates the need for this step because the signal is obtained from the SAR ADC.

*2.2. System Architecture and Operation*

Figure 4 shows a block diagram of the proposed LC-SAR hybrid ADC architecture, which consists of three parts: the LC ADC, system clock generation module CLK_GEN, and 11-bit SAR ADC. The LC ADC is a fixed-window type, and the 6-bit DAC is an input-charge scaling module that is connected to capacitor $C_6$ with the same capacitance size via switch $SW_2$. The top plate of $C_6$ goes to $V_{ref}$ via switch $SW_3$, and its output signal $V_{LCIN}$ goes to both the antiphase input terminal of comparator $COMP_1$ and the in-phase input terminal of comparator $COMP_2$. These are components of the level-crossing detection loop. The output signal CHANGE of the LC ADC is obtained from the outputs of the two comparators by performing a NAND operation. EVENT LOGIC is an event-driven logical operation module that implements the adaptive clock function of the system. The output signals $S_0$ and $S_1$ were used to control the frequency of the CLK_GEN output signal. The output signals of the CLK_GEN are the system clock and synchronization signals that control the entire system. The FS determines whether the SAR ADC performs data conversion, whereas CLK_DAC controls the LC ADC. The SAR ADC outputs an 11-bit digital signal, D[10:0], which is used directly as the output signal of the entire ADC system. The first six bits of the signal are also fed into the 6-bit DAC of the LC ADC to regulate its switching.

Figure 5 shows the operating timing diagram of the LC-SAR hybrid ADC. During the operation of the LC ADC, the sampling process occurs when the CLK_DAC signal is logically high. Conversely, when the CLK_DAC signal is at a logic low level, the LC ADC performs level-crossing detection and generates a CHANGE signal as the output. In SAR ADC, the sampling and conversion processes begin at the positive edge of the FS signal. The conversion is completed within approximately 80% of the clock cycle, and the resulting conversion result is outputted by the SAR ADC. The CLK_DAC signal is at a logic high

level before time t − 1, and the input signal Vin is connected to the 6-bit DAC through the Bootstrap switch. During this period, an input-charge scaling operation occurs. Figure 4 illustrates the switching control approach employed by the 6-bit DAC. When CLK_DAC is at a logic high level, the backplanes of the 6-bit DAC are connected to $V_{ref}$ or $V_{SS}$ based on the high six-bit output signals D[5]–D[10] of the 11-bit SAR ADC. According to the timing diagram shown in Figure 5, the SAR ADC must complete the conversion within a single cycle. This is because there is a conversion delay in the process and the result of the previous SAR ADC conversion must be available for the LC ADC to begin operating in the following cycle. Figure 5 illustrates that at time t − 1, following the arrival of the FS edge, the SAR ADC promptly completes the sampling process and starts the conversion. At time t, the SAR ADC completes the conversion and provides the conversion result. In addition, the 6-bit DAC changed the level (either $V_{ref}$ or $V_{SS}$) received by the base plate based on this result. The digital code produced by the SAR ADC at this moment can be estimated as the input signal at the previous moment, Vin(t − 1). At time t, following the output conversion result of the SAR ADC, the 6-bit DAC completes the reconstruction in the remaining one-fifth of the clock cycle. At this point, the signal input to the LC ADC is Vin(t). When the CLK_DAC switches from a high logic level to a low logic level, all the components on the bottom planes of the 6-bit DAC are connected to the $V_{SS}$. At this point, the output $V_{dac}$ of the top planes of the 6-bit DAC can be estimated.

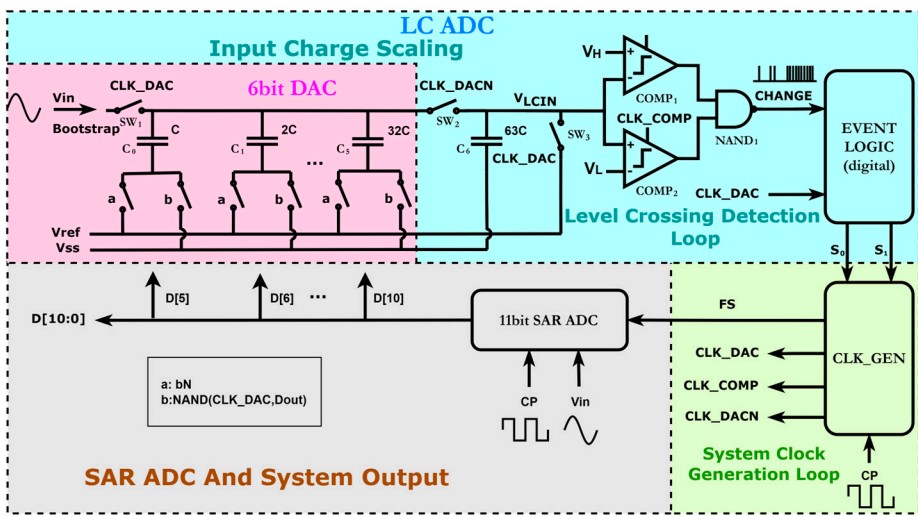

**Figure 4.** Block diagram for the proposed LC-SAR hybrid ADC architecture.

$$V_{dac} = Vin(t) - Vin(t-1) \tag{4}$$

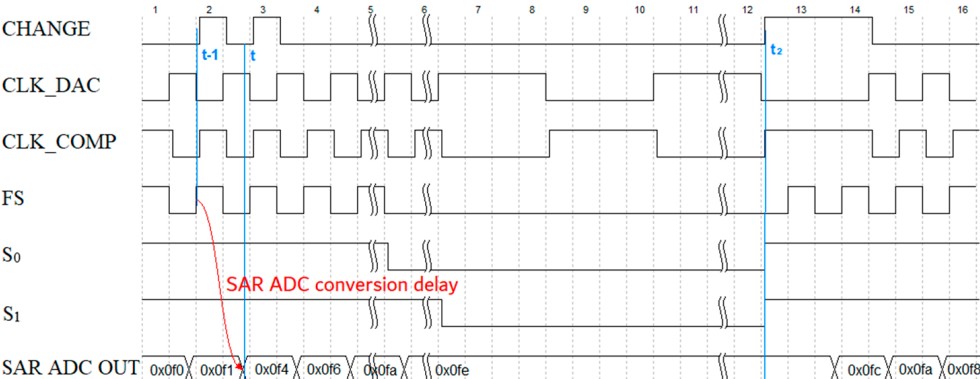

**Figure 5.** System timing diagram.

The total charge $Q_{dac}$ stored on the 6-bit DAC is

$$Q_{dac} = 63C * V_{dac} \tag{5}$$

At the same time, $SW_2$ closes and the 6-bit DAC is connected to $C_6$; before this, the total charge on $C_6$ is $Q_{C_6} = 63C * V_{ref}$, but after $SW_2$ closes, the 6-bit DAC is redistributed with $C_6$ and finally $V_{LCIN}$ is obtained as

$$V_{LCIN} = \frac{(Q_{dac} + Q_{C_6})}{(C_{dac} + C_{C_6})} = 63C * \frac{\left(V_{dac} + V_{ref}\right)}{63C + 63C} \tag{6}$$

$$V_{LCIN} = \frac{\left(Vin(t) - Vin(t-1) + V_{ref}\right)}{2} \tag{7}$$

$V_{LCIN}$ is the input to comparators $COMP_1$ and $COMP_2$ for level-crossing detection. The value of $V_H$ is given by $V_{CM} + LSB$, and the value of $V_L$ is given by $V_{CM}$-LSB. The hardware precision of the LC ADC designed in this paper is 6-bit, so the value of the LSB is

$$LSB = \frac{V_{ref}}{2^{N+1}} \approx 14(\text{mV}) \tag{8}$$

From the analysis conducted, it is obvious that the designed LC ADC detects level crossings by calculating the difference between the inputs at the approximate previous moment $(t - 1)$ and the current moment $(t)$. When this difference exceeds the LSB, a level crossing is detected. LC ADC level-crossing detection does not require consideration of the rate of change of the input signal.

In addition, by analyzing the timing in Figure 5, it can be observed that in the absence of any level-crossing detection, the CHANGE signal ceases pulsing. After a few CLK_DAC cycles, $S_0$ transitions to a low level, causing the FS to no longer provide clock pulses. Consequently, the SAR ADC stops the conversion process. This can significantly reduce power consumption. If no level crossing is detected after a certain period, as indicated by the absence of pulsing in the CHANGE signal, $S_1$ will be set to a low level. Consequently, the CLK_DAC frequency was adjusted to one-fourth of its original value, resulting in an LC ADC operating at one-fourth of its original frequency. This feature enables a straightforward adaptive operating frequency, which results in power conservation when extremely infrequent signals are encountered in the temporal domain. When a pulse arrives at the CHANGE signal, such as at time $t_2$ in Figure 5, $S_0$ and $S_1$ simultaneously transition to a high state, thereby activating the entire system. This activation triggers the SAR ADC to initiate the conversion process and to restore the LC ADC to its regular working frequency.

## 3. LC-SAR Hybrid ADC Circuit Implementation

### 3.1. LC ADC

Unlike traditional designs, the SAR ADC outputs the converted digital signal directly. The primary functions of the LC ADC in the system are to perform level-crossing detection and work together with the EVENT LOGIC to regulate the sampling clock of the SAR ADC. Consequently, the precision demand of the LC ADC is low, which reduces its complexity. Given that the proposed architecture fulfills the requirement for the operating frequency of the LC ADC for the input signal rate of change, it is feasible to operate the LC ADC at lower frequencies. The combination of low operating frequency and low accuracy requirements opens up possibilities for creating LC ADCs with extremely low power consumptions. The LC ADC is designed to ensure that the sampling rate requirement of the SAR ADC for a 20 kHz bandwidth is reached. It operates at a maximum frequency of 40 kHz and has a hardware precision of six bits. Figure 6 shows a circuit schematic of the LC ADC.

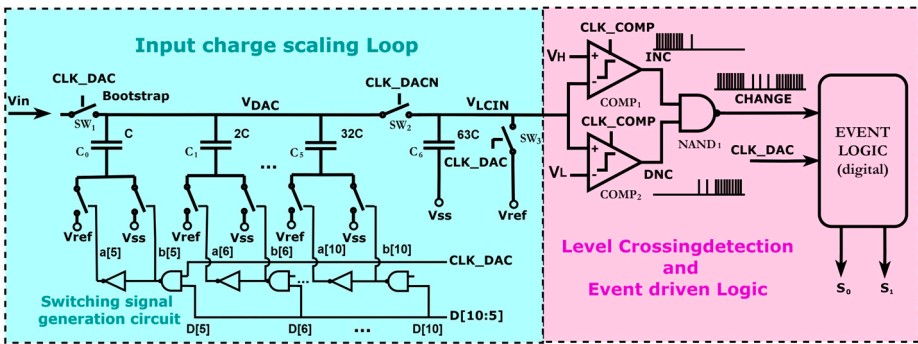

**Figure 6.** LC ADC circuit schematic.

The designed LC ADC is classified as a fixed-window type, which is popularly referred to for its lower power consumption compared with the floating-window type. Bootstrap switches are implemented at the signal inputs to maintain stable on-resistance, despite variations in the input signal amplitude. This facilitates stable signal sampling and holding, ultimately enhancing the accuracy and performance of the sampling circuit. The input charge scaling loop encompasses the switching signal generation circuit, which generates 12 switching signals by applying CLK_DAC and the upper six bits of data, D[10]–D[5], from the SAR ADC. These signals determine whether the switches are open or closed during operation of the LC ADC. To achieve an equitable division of charges across the 6-bit DAC and $C_6$ during charge sharing, the capacitance of $C_6$ was determined as the total capacitance value of $C_0$ to $C_5$. The output of the input charge scaling loop is $V_{LCIN}$, which is connected to two comparators that are applied for level-crossing detection. The LC ADC generates the output signals INC, DNC, and CHANGE. However, in this design, the INC and DNC signals need not be extracted separately and only the CHANGE signal is fed into the EVENT LOGIC. Figure 7 shows the workflow diagram of EVENT LOGIC.

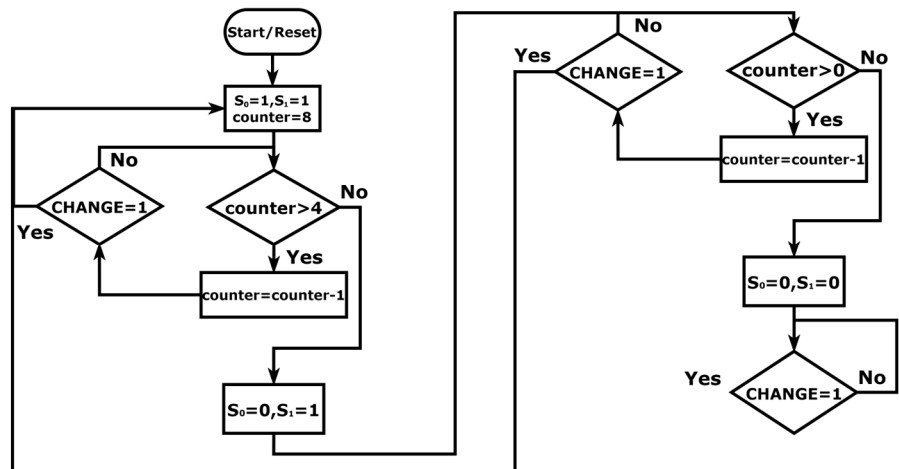

**Figure 7.** EVENT LOGIC workflow diagram.

Figure 6 illustrates the EVENT LOGIC, which receives two input signals: CHANGE and CLK_DAC. The output of the LC ADC is CHANGE, which is determined by comparing $V_{LCIN}$ with $V_H$ and $V_L$ following the NAND operation. When the CHANGE signal switches from a logic low level to a logic high level, it signifies the detection of level crossing. The CHANGE signal was used to control the operational status of EVENT LOGIC. Figure 4 illustrates that CLK_DAC is the output generated by CLK_GEN, which controls the operations of the LC ADC and functions as a clock signal for the EVENT LOGIC. Figure 7 illustrates that, following the initiation of the EVENT LOGIC reset, the two output signals $S_0$ and $S_1$ are configured as $S_0 = 1$ and $S_1 = 1$, respectively, while

the counter is set to 8. When $S_0 = 1$ and $S_1 = 1$, the LC and SAR ADC are controlled to perform detection and conversion at a standard speed. The following procedure checks whether the counter is greater than 4. If the counter decreases, then it checks whether the CHANGE signal is set to a logic high level, pointing to a level-crossing detection. If it is, the procedure jumps to the beginning, setting $S_0$ and $S_1$ to 1 and setting the internal counter to 8. If the CHANGE signal is not set to a logic high level, the procedure continues to check whether the counter is greater than 4 and the cycle is repeated. If no level crossing is detected after four CLK_DAC clock cycles, the counter is checked to determine if it exceeds four. If there are no more than 4, the next process is entered, the output control signal $S_0$ is set to 0, and $S_1$ is set to 1. This controls the LC ADC to detect at normal speed and stops converting the SAR ADC. The next process checks whether the counter value is greater than 0. If the counter decreases, then it checks whether the CHANGE signal is set to a logic high level, indicating a level-crossing detection. If it is, the procedure jumps to the beginning, sets $S_0$ and $S_1$ to 1, and sets the internal counter to 8. If the CHANGE signal is not set to a logic high level, the procedure continues to check whether the counter is greater than 0 and the cycle is repeated. If no level crossing is detected after four CLK_DAC clock cycles, the counter is checked to verify if it is greater than 0. If the result is NO, the next step is entered and the output control signals $S_0$ and $S_1$ are set to 0. The LC ADC was controlled to operate at a reduced speed of one-fourth of the usual speed, whereas the SAR ADC maintained its conversion stop. The subsequent procedure establishes whether the CHANGE signal is set to a logic high level, indicating the detection of level crossing. If this condition is satisfied, the program proceeds by setting $S_0$ and $S_1$ to 1 and initializing the internal counter to a value of 8. Conversely, if the condition is not met, the program continues to repeat the evaluation and waits for the CHANGE signal to transition to a logically high level.

A circuit schematic of the comparator utilized for the LC ADC in the proposed LC-SAR hybrid ADC is illustrated in Figure 8. The LC ADC precision in this study is specified as 6 bits, which significantly relaxes the requirements for the comparator's offset voltage and noise performance. A two-stage comparator structure was selected; the initial stage consisted of a dynamic amplifier, whereas the subsequent stage was a strong-ARM latch. In contrast to the single-stage strong-ARM dynamic comparator [22], the two-stage configuration exhibited an enhanced speed and precision. $INV_1$ and $INV_2$ were implemented in the front and rear stages, respectively, which can reduce kickback noise. Figure 9 shows the outcomes of 1000 Monte Carlo simulations pertaining to the offset voltage of the comparator.

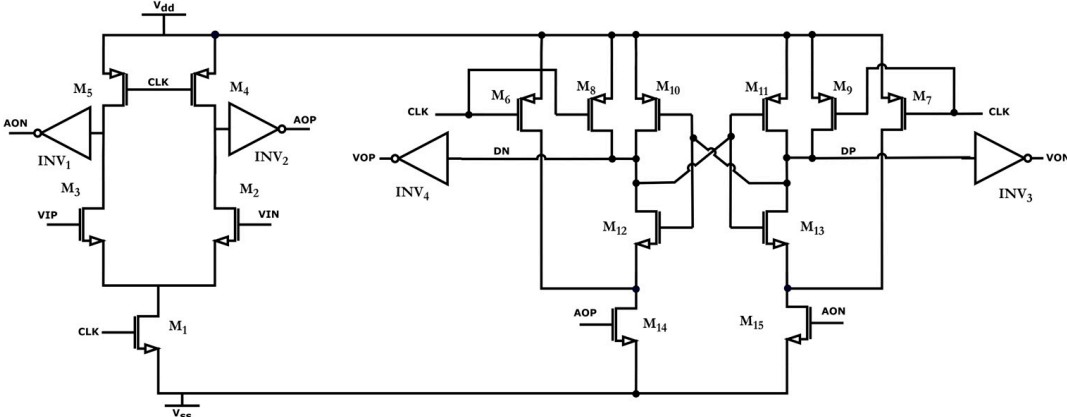

**Figure 8.** The comparator circuit of LC ADC.

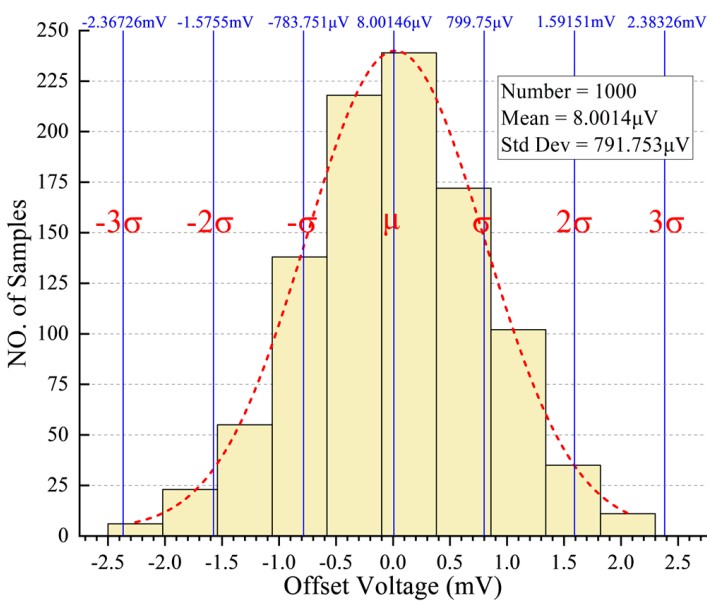

**Figure 9.** A total of 1000 offset voltage Monte Carlo simulations of the comparator.

Monte Carlo simulation results often use the mean ($\mu$) plus three times the standard deviation ($3\sigma$) as the final result, following the three-sigma rule. This indicates that approximately 68% of the data fall within one standard deviation of the mean, approximately 95% of the data fall within two standard deviations of the mean, and approximately 99.7% of the data fall within three standard deviations of the mean. This means that if one has a normal distribution, the empirical rule can be used to estimate the percentage of data that fall within a certain range [23]. The offset voltage of the comparator was 2.38326 mV. When a reference voltage of 1.8 V was applied to a 6-bit LC ADC, the LSB size was 14 mV. In accordance with the system design specifications, the comparator has an offset voltage below LSB/2.

### 3.2. CLK_GEN

As shown in Figure 4, the LC ADC outputs signals $S_0$ and $S_1$ to CLK_GEN and CLK_GEN generates clock signals based on $S_0$ and $S_1$. Thus, the following timing requirements exist for an LC ADC.

1.  A maximum operating frequency of 40 kHz;
2.  When $S_0$ = 1 and $S_1$ = 0, the LC ADC operates at one-fourth of the original normal operating frequency (10 kHz).

    For SAR ADCs:

1.  A sampling frequency of 40 kHz;
2.  Stops working when $S_0$ = 0 and $S_1$ = 1, i.e., FS no longer outputs pulses;
3.  The SAR ADC requires 13 clock cycles to complete a conversion.

The CLK_GEN circuit is shown in Figure 10. The CP frequency was set as 520 kHz to satisfy the timing requirements of the SAR ADC. This frequency is divided by 13 and a 40 kHz frequency CLK_DIV$_1$ is generated that is used to drive both the LC ADC and the SAR ADC. The CLK_DIV$_1$ signal then goes through a 4-division, which consists of two D flip-flops, resulting in a CLK_DIV$_2$ signal with a frequency of 10 kHz. The CLK_DAC was used to provide the necessary driving signal for the operation of the LC ADC. The $S_1$ signal reaches the D terminal of DFF$_3$, guaranteeing the phase continuity for CLK_DAC. This is applied to regulate CLK_DAC output, either CLK_DIV$_1$ or CLK_DIV$_2$. CLK_COMP is generated from CLK_DAC using a delay buffer and inverter INV$_8$. Signal $S_0$ is connected to the S-terminal of the second multiplexer MUX$_2$ to regulate the FS. When the logic level of $S_0$ is high, the FS outputs CLK_DIV$_1$, which causes the SAR ADC to operate at a sampling

rate of 40 kHz. Conversely, when the logic level of $S_0$ is low, the FS ceases to output a clock signal, which causes the SAR ADC to stop operating. This effectively reduces the purpose of reducing power consumption. According to Figure 7, when level crossing is detected, CHANGE switches from a low logic level to a high logic level. In addition, the EVENT LOGIC generates outputs $S_0 = 1$ and $S_1 = 1$. At this time, the LC and SAR ADCs are controlled to detect and convert at a standard speed. This also implies that CLK_DAC and the FS generate clock signals with a frequency of 40 kHz. According to Figure 7, if level crossing is detected, the logic level of CHANGE changes from low to high. EVENT LOGIC outputs $S_0 = 1$ and $S_1 = 1$, which control the LC ADC and SAR ADC to detect and convert at normal speed. This implies that CLK_DAC and the FS output a signal with a frequency of 40 kHz. When $S_0 = 0$ and $S_1 = 1$, the LC ADC is detected at a normal speed, whereas the SAR ADC stops the conversion. This means that the FS no longer outputs a clock signal but CLK_DAC continues to output a clock signal with a frequency of 40 kHz. Finally, when $S_0 = 0$ and $S_1 = 0$, the LC ADC is detected at a quarter of the normal speed and the SAR ADC stops converting. This causes the CLK_DAC to output a clock signal with a frequency of 10 kHz, whereas the FS does not output any clock signal. By utilizing this mode of operation, the system can operate with minimal power consumption when the input signal frequency is exceedingly low.

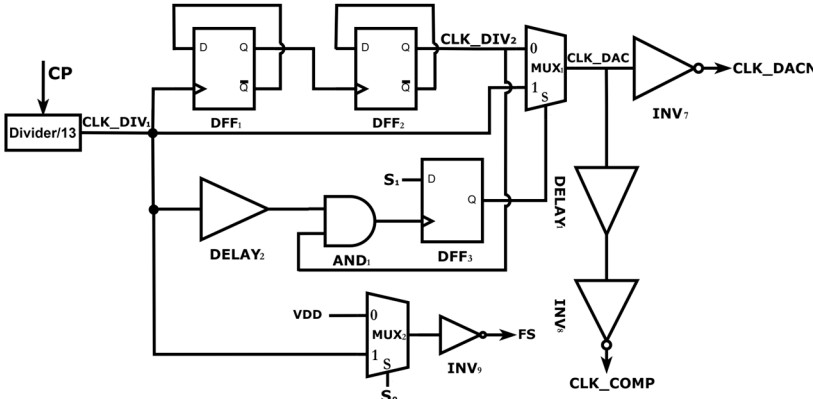

**Figure 10.** CLK_GEN circuit schematic.

### 3.3. SAR ADC

A differential structure was adopted in the SAR ADC; its single-ended model is shown in Figure 11. To improve the performance of the sampling circuit, the input signal was connected to the CDAC array via a bootstrap switch.

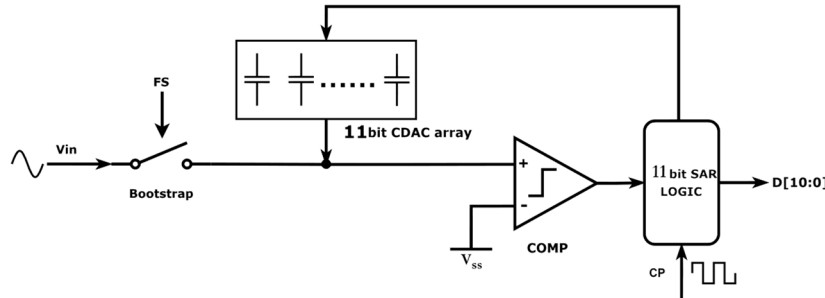

**Figure 11.** SAR ADC circuit block diagram.

To achieve low power consumption, the SAR ADC operates at the Nyquist bandwidth rather than at the oversampling frequency. To satisfy the 20 kHz bandwidth requirement, the sampling frequency of the SAR ADC was set to 40 kHz. This ensured that the acquired signals were not subject to aliasing.

The challenge in SAR ADC design involves designing a CDAC array. The architecture of the CDAC capacitor array used in this study is shown in Figure 12. It has a segmented capacitor architecture, which offers a higher capacitance-saving capability than the non-segmented capacitor architecture. The working principle of this charge redistribution structure capacitor array is described in Ref. [24].

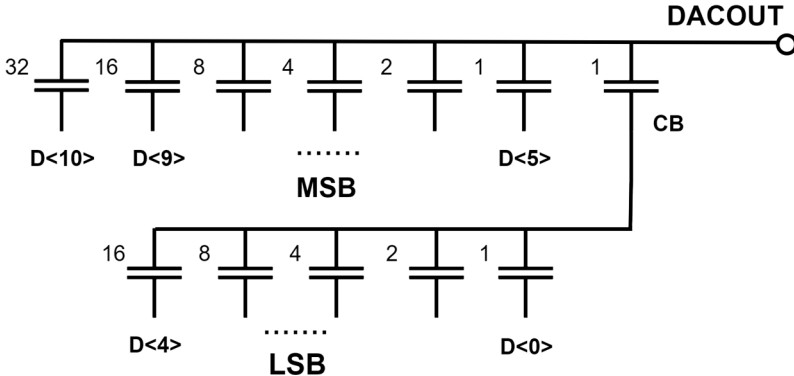

**Figure 12.** Segmented CDAC arrays used in SAR ADC design.

Choosing the appropriate size of the unit capacitance C is key to ensuring that the circuit is not affected by the negative effects of the capacitor mismatch after physical implementation. Here, it is assumed that the unit capacitance C follows a normal distribution, with mean value $C_0$ and standard deviation $\sigma_c$. Therefore, with capacitors connected in parallel, the total capacitance size is $mC_0 + m^{1/2}\sigma_c$. For an N-bit converter, to reduce the error caused by the capacitance mismatch to less than LSB/2, the following equation [25] should be satisfied:

$$\left| \frac{C_{msb}}{C_{total}} - \frac{1}{2} \right| < \frac{1}{2^{N+1}} \tag{9}$$

For an N-bits segmented capacitor array with M-bits high segmentation, $C_{msb}$ and $C_{total}$ are, respectively,

$$C_{msb} = 2^{M-1}C_0 \pm 2^{\frac{M-1}{2}}\sigma_c \tag{10}$$

$$C_{total} = 2^{M-1}C_0 \pm 2^{\frac{M-1}{2}}\sigma_c + 2^{M-1}C_0 \pm 2^{\frac{M-1}{2}}\sigma_c \tag{11}$$

Therefore, by combining with Equation (9), we can determine that the unit capacitance size of the segmented capacitor array should satisfy Equation (12).

$$\left| \frac{2^{M-1}C_0 \pm 2^{\frac{M-1}{2}}\sigma_c}{2^{M-1}C_0 \pm 2^{\frac{M-1}{2}}\sigma_c + 2^{M-1}C_0 \pm 2^{\frac{M-1}{2}}\sigma_c} - \frac{1}{2} \right| < \frac{1}{2^{N+1}} \tag{12}$$

When the standard deviation of the capacitance of the highest bit is positive and that of the rest of the capacitance is negative, Equation (13) can be derived from the worst-case scenario.

$$\frac{2^{\frac{M-1}{2}}}{2^M} \times \frac{\sigma_c}{C_0} < \frac{1}{2^{N+1}} \tag{13}$$

The standard deviation of the capacitance followed a normal distribution. To enhance the reliability of the converter in this study, the three-sigma rule of the normal distribution was utilized to determine the unit capacitance size C. By using the equation $\varepsilon = \sigma_c/C_0$ and combining it with Equation (13) and the three-sigma rule, it can be determined that the size of the segmented capacitance must meet the following equation

$$3\varepsilon < \frac{1}{2^{\left[\frac{2N-M+1}{2}\right]}} \tag{14}$$

The converter in Figure 12 utilizes a capacitor array with 6 bits for high segmentation and 5 bits for low segmentation. This results in N = 11 and M = 6. The obtained value of ε is less than 0.092%. Table 1 lists the values of the standard deviation of C for various capacitance values during the process.

**Table 1.** The standard deviation of the unit capacitance for various capacitance values in the present process.

| Mean(F) | 20.2825 f | 112.201 f | 429.389 f | 1.28815 p | 1.71752 p |
|---|---|---|---|---|---|
| Std Dev(F) | 91.3447 a | 202.147 a | 386.818 a | 669.989 a | 773.647 a |
| ε | 0.45036% | 0.180165% | 0.0900856% | 0.052011% | 0.045044% |

It is evident that the unit capacitance C of 429.389 fF satisfies the given criteria. If one selects the "6 + 6" segmented 12-bit precision CDAC array, it must satisfy the ε < 0.046% requirement. According to Table 1, the unit capacitance C must be chosen to be 1.71752 pF, which is significantly large. This results in not only a large chip area but also in high power consumption. The digital calibration technology of the capacitor can reduce the chip area to a certain extent, but a significant increase in power consumption occurs. This is why the 11-bit precision CDAC array was designed in this study, achieving a favorable tradeoff between area and power consumption.

In addition to the capacitor array, comparator noise and offset voltage are additional variables that affect the converter precision. Figure 13 illustrates the dynamic comparator employed in the SAR ADC. In the case of constant offset voltage, the dynamic efficacy of the converter, including SNDR, remains unaffected. The offset voltage of the comparator should be maintained as low as possible. If it varies with the input signal, it has a significant impact on the overall performance of the converter. The preamplifier stage was represented by transistors $M_0$–$M_6$. In contrast to the conventional preamplifier structure, the comparator integrates two cross-coupled NMOS transistors ($M_1$ and $M_2$) situated between the tail-current transistor ($M_0$) and input-pair transistors ($M_3$ and $M_4$). This configuration increases the pre-amplification stage gain, facilitates charge reuse, and decreases power consumption. The dynamic latch stage comprises transistors $M_7$–$M_{19}$. In contrast to conventional dynamic latches, transistors $M_{13}$ and $M_{14}$ consist of two NMOS transistors that substitute the initial PMOS tail-current transistors. This configuration permits the potentials at the M and N points to decrease to $V_{DD}$–$V_{THN}$ ($V_{THN}$ is the threshold voltage of the NMOS transistors) when CLK is eliminated. As a result, the comparator was capable of retaining a threshold voltage change for every comparison. To decrease the offset, a large W/L was specified on the input pair of transistors ($M_3$ and $M_4$) of this comparator.

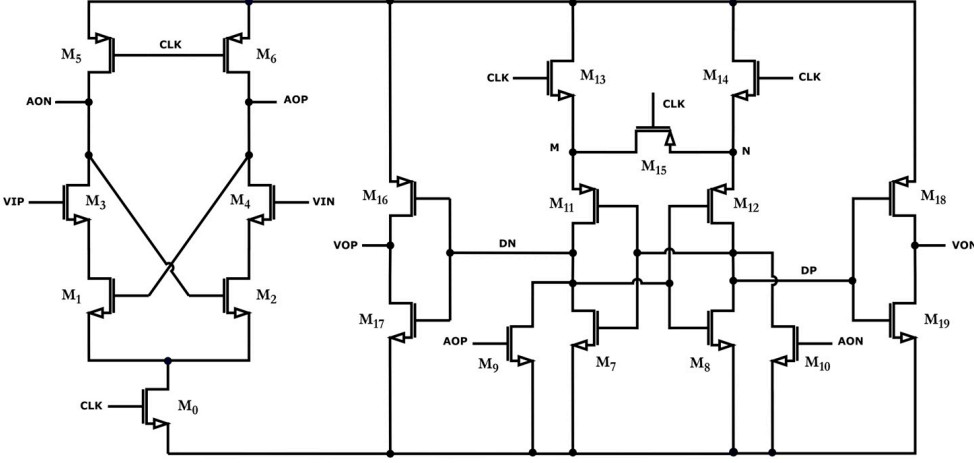

**Figure 13.** Low offset low power dynamic comparator circuit.

The 11-bit SAR ADC has an LSB of 878.9 μV. Consequently, the comparator offset voltage should not exceed 439.45 μV (LSB/2). A total of 1000 Monte Carlo simulation results are shown in Figure 14. As can be seen, the comparator's offset voltage −308.8 V satisfies the requirement. The process corner simulation outcomes are illustrated in Figure 15, which shows that the offset voltage of the comparator fluctuates between 0 V and −180 μV. This value aligns with the results obtained from the Monte Carlo simulation and fulfills the design criteria for the SAR ADC.

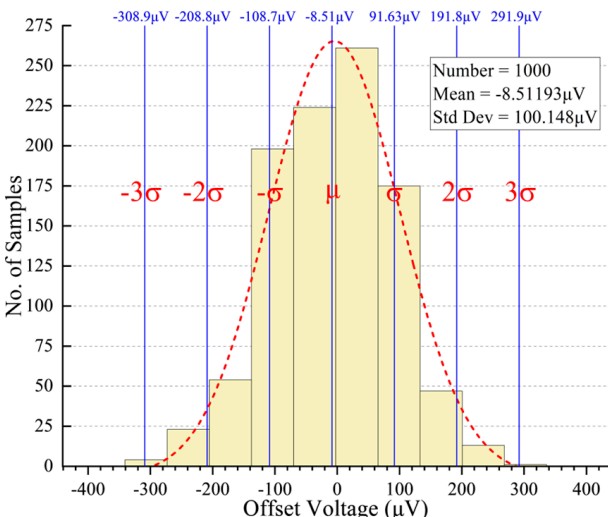

**Figure 14.** Results of 1000 Monte Carlo simulations of the designed comparator with offset voltage.

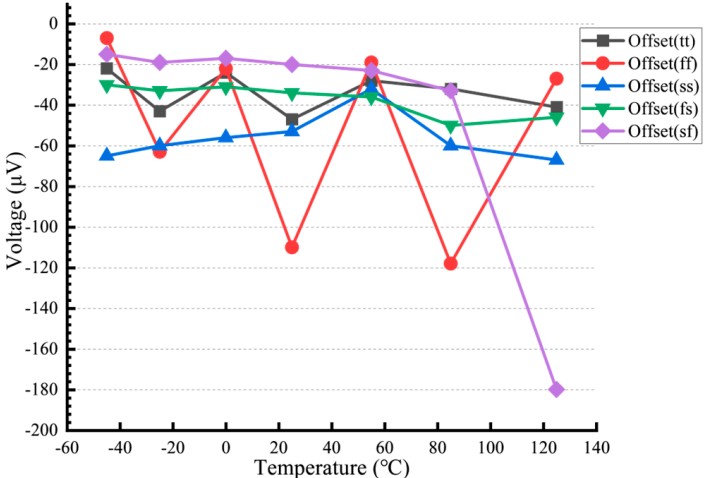

**Figure 15.** Simulation results of the comparator with different process corners for offset voltage.

## 4. Simulation Results and Analysis

An LC-SAR hybrid ADC was designed and verified using a 180 nm CMOS process. The ADC occupies an area of 1.167 mm$^2$, including the ESD and bonding pads, as shown in Figure 16. The chip is designed to operate at a supply voltage of 1.8 V. Some of the digital circuits within the chip operated at a lower supply voltage of 1 V. These circuits could handle a full-scale bandwidth of 20 kHz. Furthermore, the power consumption of the device decreases as the frequency of the input signal decreases. The CLK pin of the chip requires an input clock frequency of 520 kHz, which is 13 times that of 40 kHz. VINN and VINP function as differential signal inputs, whereas the VL and VH ports are utilized as the input of the fixed window level of the LC ADC, typically $V_{CM} \pm$ LSB (where the LSB size is $1.8/2^7$, corresponding to the LSB size of a 6-bit LC ADC).

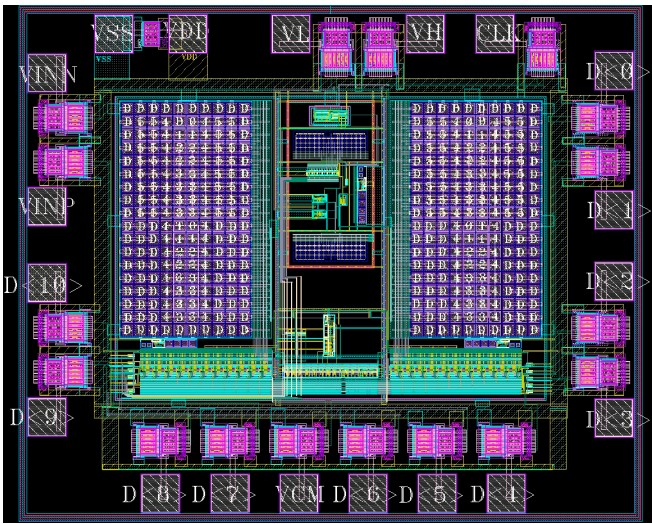

**Figure 16.** LC-SAR hybrid ADC layout.

The spectral outcomes derived from the FFT operation with 8192 data points at an input signal frequency of 14,975.5859375 Hz (approximately 14.975 kHz) and 1.8 $V_{PP}$ are shown in Figure 17. The spectrum results indicate that at a frequency of 14.975 kHz, it is possible to attain an SNDR of 67.11 dB and an SFDR of 83.55 dB with an ENOB of 10.85 bits, as well as a power consumption of 4.34 µW. The power distribution for a sinusoidal signal input at 1.8 $V_{PP}$, 14.975 kHz is shown in Figure 18a. Specifically, the 11-bit SAR ADC consumes 3.92 µW or 90.11% of the system's total power consumption. The LC ADC operates at a modest power consumption of 0.36 µW, constituting a mere 8.28% of the overall power consumption of the system. This is because of its lower operating frequency; the proposed architecture enables it to function at lower operating frequencies while disregarding the input signal rate of the change. CLK_GEN is a digital circuit that operates on a 1 V supply voltage and consumes 0.07 µW of power, or 1.61%.

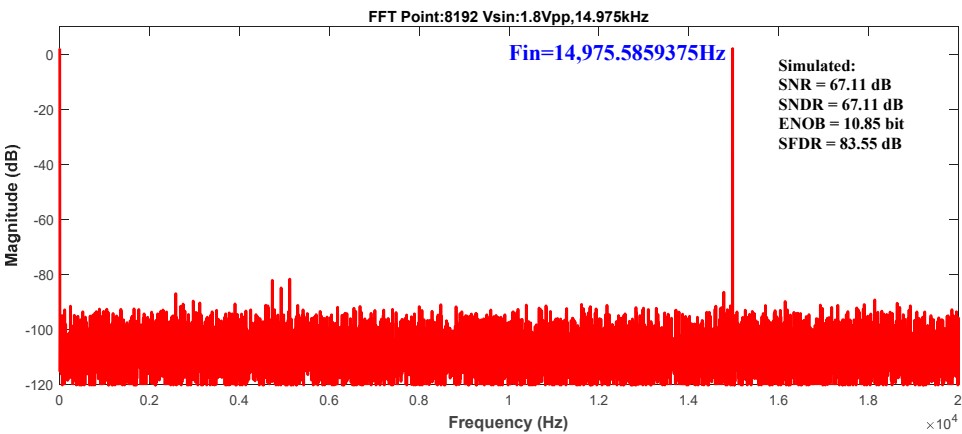

**Figure 17.** Output spectrum of ADC at 1.8 $V_{PP}$, 14.975 kHz sinusoidal signal input.

As shown in Figure 19a, the power consumption of the system and the spectrum of the output signal were evaluated within the bandwidth specified in this paper for a sinusoidal signal input with an amplitude of 1.8 $V_{PP}$ and different frequencies. The SNDR demonstrated a positive correlation with the frequency when the signal frequency fell below 0.2 kHz. This is because a lower frequency corresponds to a lower signal integrity captured, necessitating signal interpolation for FFT data acquisition, which subsequently results in a reduction in SNDR. An input sinusoidal signal frequency of 50 Hz results in an SNDR of 58.77 dB, as demonstrated by the results. EVENT LOGIC can direct the SAR

ADC to conduct comprehensive sampling and maintain a high level of signal integrity when the signal frequency exceeds 0.2 kHz. When the input sinusoidal signal frequency was set to 1 kHz, the SNDR reached a maximum of 67.41 dB. Owing to the event-driven logic algorithm described above, the power performance of the ADC was optimized over the entire frequency band. The power consumption decreased as the frequency decreased, provided that the signal frequency was less than 1 kHz. The system has a power consumption of 2.08 μW at an input sinusoidal signal frequency of 50 Hz. The power consumption exceeds 4.3 μW when the frequency surpassed 1 kHz, mostly because of the uninterrupted operation of the SAR ADC. The power consumption of each module in the ADC depends on the frequency of the input signal, as illustrated in Figure 19b. The minimum power consumption occurs when the ECG signal is input. For input signals with a frequency below 1 kHz, there is a direct proportional relationship between the power consumption and the input signal. The power consumption of each module remained relatively constant for the input signals with frequencies above 1 kHz.

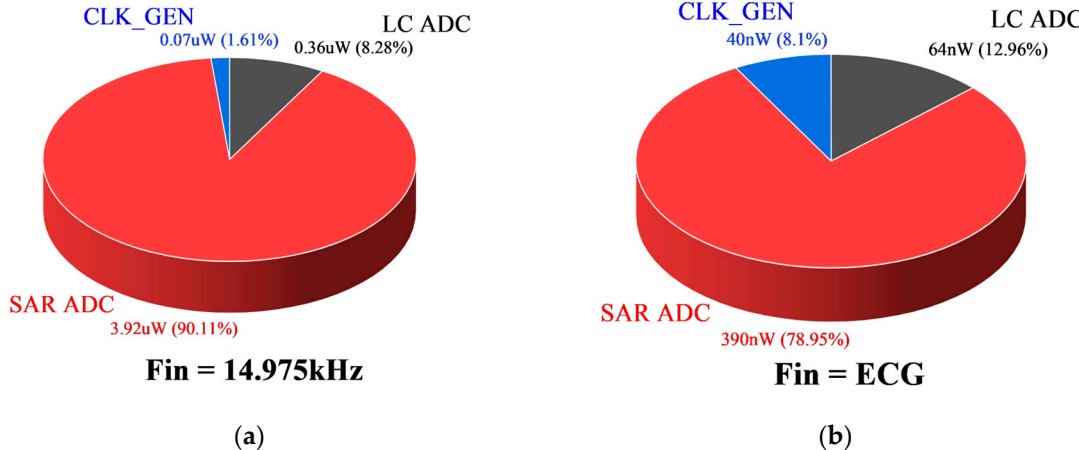

**Figure 18.** (**a**) Power consumption distribution with 14.975 kHz sinusoidal input signal. (**b**) Power consumption distribution with input ECG signal.

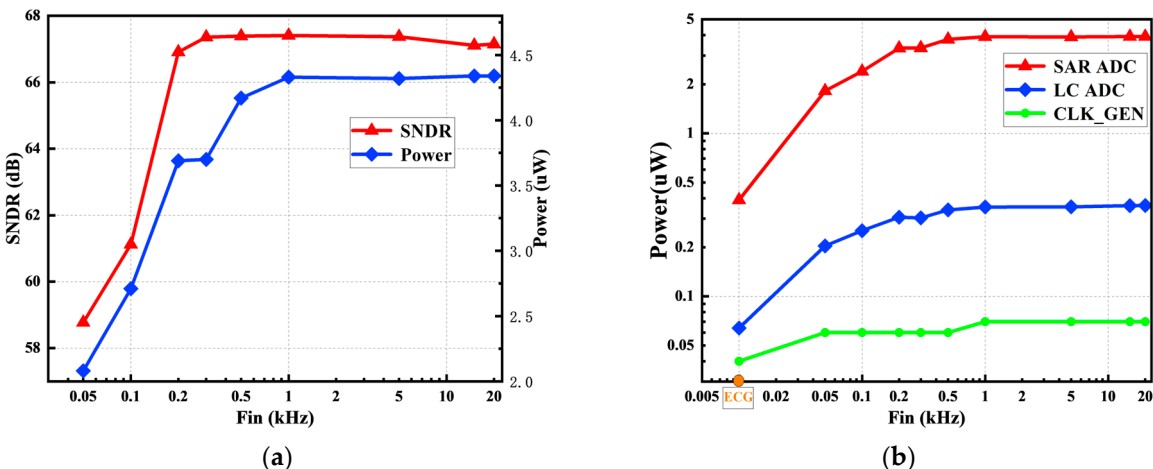

**Figure 19.** (**a**) SNDR and power of ADC with different frequency input signals. (**b**) Power of each module under different frequency input signals.

The results of the process corner simulations for the proposed converter at temperatures ranging from −45 °C to 125 °C are depicted in Figure 20.

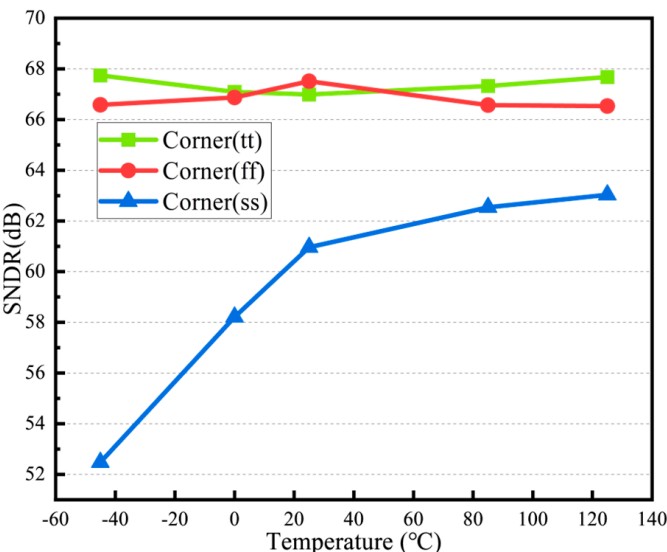

**Figure 20.** SNDR simulation results for different process corners.

Figure 21 shows the simulation results of the system output when a sinusoidal signal with a frequency of 200 Hz and amplitude of 1.8 $V_{PP}$ is used as the converter's input signal. VINN is the input signal of the LC ADC and CHANGE represents its output signal. $V_{DAC}$ is the output of the capacitive DAC top plate used in the LC ADC, as shown in Figure 6. Vout denotes the SAR ADC output. $S_0$ and $S_1$ are the output signals of the LC ADC that are used to control CLK_GEN to generate the system clock signal. FS is the signal generated by the CLK_GEN, which controls the SAR ADC operation. Figure 21 depicts the results of a simulation segment at the peak of the input sinusoidal signal, where the rate of change of the input signal is the slowest. The simulation results show that the rate of change of the input signal slows as it approaches the crest of the wave, resulting in a decrease in the pulse frequency in the CHANGE signal. This means that the LC ADC does not detect level crossing. Thus, $S_0$ and $S_1$ send a message causing the FS to turn off the SAR ADC. Ultimately, the power consumption of the system is reduced. During the 1200 µs–1600 µs time interval in Figure 21, the FS ceases emitting pulses, which means that the SAR ADC stops operating because the change in the input signal VINN is not noticeable at this time. This approach provides a high power efficiency for the converter.

The simulation results for the LC ADC output signal shown in Figure 21 are shown in Figure 22. The level reference signals, $V_H$ and $V_L$, were 914 mV and 886 mV, respectively (i.e., $V_{CM} \pm$ LSB). Level crossings were detected at x1–x6. The observed $V_{DAC}$ frequency at x4 was one-fourth of the usual operating frequency. This is because, when the input signal reaches its peak, the rate of change of the input signal decreases. This causes the LC ADC to operate at one-fourth of its normal operating frequency to reduce system power consumption. However, after detecting the level crossing, it can be seen that the LC ADC returns to its normal operating frequency to ensure that the input signal is not lost.

Figure 23 shows a 900 ms segment of the ECG input signal and the ADC output signal. The apparent ADC was only 0.49 µW. To achieve the same accuracy, the traditional SAR ADC using the uniform sampling method requires a power consumption of 3.92 µW. However, the proposed ADC can reduce power consumption by approximately 80%. Ref. [26] discussed the compression ratio (CR), which quantifies the efficiency of an LC ADC. It is defined as the ratio of the average number of bits/s in the signal obtained from a uniform sampling ADC to the average number of bits/s in the signal obtained from an LC-ADC. The amount of data captured per second by the LC-SAR hybrid ADC cannot be explicitly determined for a random ECG signal but can be estimated by observing the power

consumption over a period of time. The conventional uniform sampling ADC requires 3.92 μW to acquire a 900 ms ECG signal, whereas the proposed level-crossing-sampling ADC consumes only 0.49 μW. The average rate of data acquisition per second was estimated based on the power consumption. Thus, the CR was approximately eight times. The power consumption ratio of each module in the system is shown in Figure 18b. The SAR ADC consumes a mere 390 nW, which corresponds to 78.95% of the total power consumption. The LC ADC consumes 64 nW, accounting for 12.96% of the total power consumption. Finally, CLK_GEN consumes 40 nW, representing 8.1% of the total power consumption.

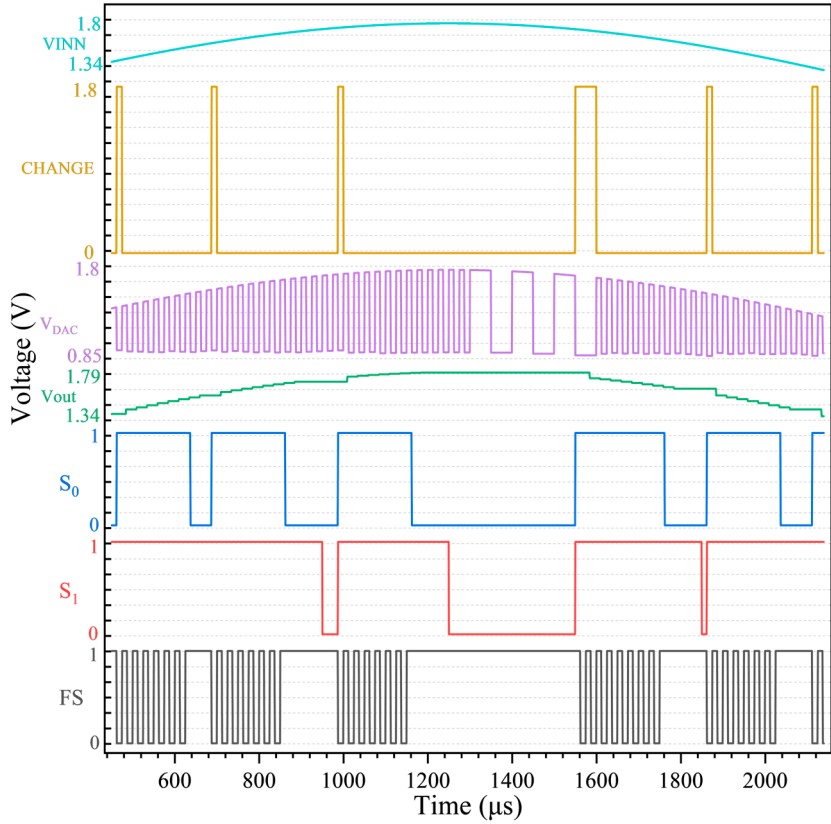

**Figure 21.** Transient simulation of the proposed ADC with a sinusoidal signal input of 1.8 $V_{PP}$, 200 Hz.

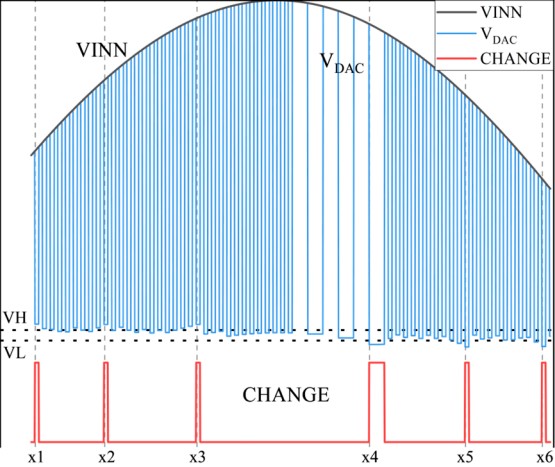

**Figure 22.** Simulation results of level-crossing detection with sinusoidal signal input at 1.8 $V_{PP}$, 200 Hz.

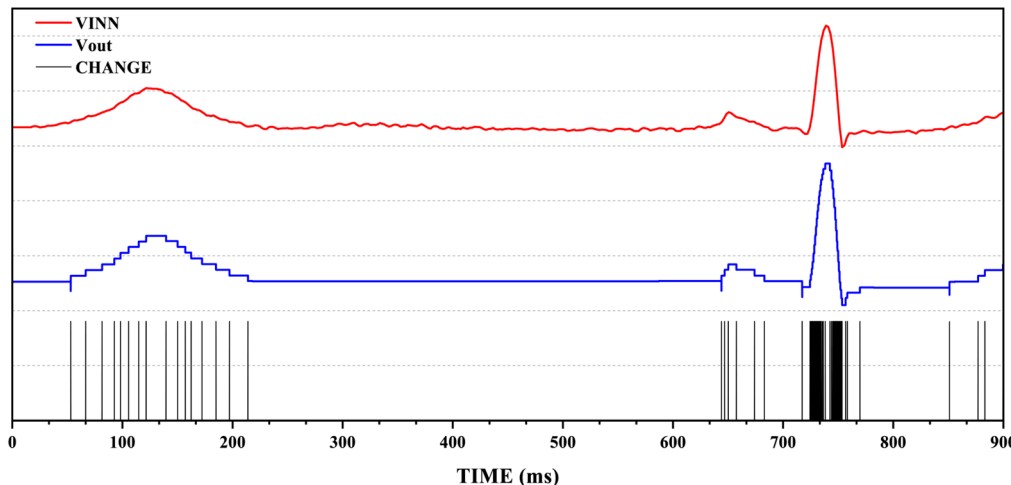

**Figure 23.** ECG signal input and ADC output.

The performance of the proposed LC-SAR hybrid ADC is summarized in Table 2, which provides a comparison with the most recent global advancements. The combined design of an 11-bit SAR ADC and 6-bit LC ADC in this study resulted in a more substantial architectural framework, with the SAR ADC covering 70% of the total area. The FoMs and FoMw were 164 dB and 58.8 fJ/conv.-steps, respectively, which are superior to those of several other modern facility architectures. The 10.85-bit ENOB and 0.49~4.34 μW power consumption exhibit a good tradeoff between power efficiency and precision.

**Table 2.** Comparison of performances the proposed and conventional ADCs.

| Parameter | [7] | [9] * | [11] * | [10] * | [17] | This Work |
|---|---|---|---|---|---|---|
| Topology | LC | LC | SAR | LC-SAR | LC | LC-SAR |
| Technology (nm) | 180 | 40 | 130 | 55 | 350 | 180 |
| Active Area (mm$^2$) | - | 0.012 | 0.03 | 0.063 | - | 1.167 |
| Need External CLK | NO | YES | YES | NO | YES | YES |
| Supply Voltage (V) | 0.8 | 0.5/1 | 0.6 | 1.2 | 1/2.5 | 1/1.8 |
| Quantizer Resolution (bit) | 8 | 8 | 12 | 4 + 4 | - | 6 + 11 |
| Bandwidth (Hz) | 5~3.3k | 15k | 5k | 25k | 20k | 20k |
| Power (μW) | 0.062~0.106 | 0.92~5.38 | 1.88 | 0.44~5.7 | 30 | 0.49~4.34 |
| ENOB (bit) | 6.39 | 10.4 | 10.72 | 10.04 | 8.29 | 10.85 |
| SNDR (dB) | 40.2 | 64.4 | 66.3 | 62.2 | 39.5 | 67.41 |
| FoMw (fJ/conv.-step) | 132.5~191.7 | 132.7 | 111.4 | 29.7~108.3 | 2396 | 58.8 |
| FoMs (dB) | 119.2~145.1 | 158.8 | 160.5 | 158.6 | 127.7 | 164 |

FoMw = Power/(2*BW*2$^{ENOB}$); FoMs = SNDR + 10log(BW/Power); * prototype fabricated in silicon.

## 5. Conclusions

This paper proposes a high-energy-efficiency LC-SAR hybrid ADC for spare IoT data sampling applications that combines the ultralow power consumption of an LC ADC with the relatively high conversion precision of an SAR ADC. The LC-SAR hybrid ADC incorporates the previous SAR ADC conversion result into the charge-scaling DAC of the LC ADC at the current time and uses the difference between the input signal at the previous time and the current time to detect level crossing. This allows level-crossing detection by comparing the input signals from the prior and present moments, which allows the LC ADC to operate at a lower sampling frequency and power consumption even when the input signal change rate is high. The simulation findings indicate that the architecture proposed in this study has lower power consumption while maintaining the same bandwidth and accuracy. The LC-SAR hybrid ADC was constructed utilizing a CMOS 180 nm process. The simulation results indicate that the ADC can attain an SNDR

of 67.41 dB and an SFDR of 83.55 dB over a 20 kHz bandwidth. Furthermore, it supports an ENOB of up to 10.85 bits and consumes a maximum of 4.34 µW of power. In the case of an ECG input signal, the power consumption is only 0.49 µW, representing an over 80% reduction in comparison to the conventional uniform sampling ADC with respect to the number of sampling points. The designed ADC is suitable for IoT applications that require extremely low power consumption such as mobile medical diagnostic devices.

**Author Contributions:** Conceptualization, W.X.; Methodology, W.X. and H.T.; Software, H.T. and W.X.; Validation, H.T.; Formal analysis, W.X., H.T. and B.W.; Investigation, W.X., H.T., X.W. and H.L.; Resources, W.X., H.T., X.W., B.W. and H.L.; Data Curation, W.X. and H.T.; Writing—original draft, H.T.; Writing—Review and Editing, W.X. and H.T.; Visualization, W.X. and H.T.; Supervision, W.X.; Project administration, W.X.; Funding acquisition, W.X., X.W. and B.W. All authors have read and agreed to the published version of the manuscript.

**Funding:** This research was funded in part by the National Natural Science Foundation of China (grant numbers 62064002, 62164003, and 62364009) and in part by the Guangxi Key Laboratory of Precision Navigation Technology and Application, Guilin University of Electronic Technology under grant DH202212.

**Data Availability Statement:** The data present in this study are available on request from the corresponding author.

**Conflicts of Interest:** The authors declare no conflicts of interest.

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
