# Peer review of "A 0.49–4.34 μW LC-SAR Hybrid ADC with a 10.85-Bit ENOB and 20 KS/s Bandwidth"

_electronics, doi:10.3390/electronics13061078_

Round 1
Reviewer 1 Report
Comments and Suggestions for Authors
This paper proposes a 0.49–4.34-μW LC-SAR hybrid ADC with 10.85-bit ENOB and 20 kS/s bandwidth, which performs with a good figure of merit, with FoMw and FoMs reaching 58.8 fJ/conv.steps and 164 dB, respectively. Compared with traditional LC ADC or SAR ADC, the proposed hybrid ADC combines the ultralow-power advantage of LC ADC with the high-precision advantage of SAR ADC in converting IoT data with sparse characteristics such as ECG, EEG, and brain potential. However, the dominant contribution is not clear and some errors are listed in the submitted manuscript. So I cannot recommend accepting this manuscript.
1. At the end of the Abstract, what is the meaning of the sentence “Approximately 80% of the data compression can be achieved in physiological signal acquisition applications”?
2. In Introduction, there is no reference from line 32 of page 1 to line 68 of page 2. Why?
3. The word styles of “Figure 1a-b” and “Figure 1b” do not match the template of this Journal. Please correct them with Fig. 1(a) or Fig. 1(b).
4. In Eq. (3), what is FSLCADC?
5. In Eq. (9), why do you use (3*epsilon)?
6. In the comparison of performance, why does the largest chip area of 1.167 mm2?
7. The performance comparison shows that this study is novel. However, the authors need to provide the measured results. I need to make sure that the proposed LC-SAR works correctly.
Comments on the Quality of English LanguageIt is fine for the English language.
Author Response
Mar. 5, 2024
Dear Reviewer 1,
Please find submitted the revised version of manuscript “electronics-2882271”, entitled “A 0.49–4.34-μW LC-SAR Hybrid ADC With 10.85-bit ENOB and 20 KS/s
Bandwidth”, authored by Hai Tang, Haiou Li, Baolin Wei, Xueming Wei and me.
We have incorporated the Editor’s and referees’ recommendation into the revised manuscript. We highly appreciate the valuable time and pertinent remarks and suggestions from the reviewers, which have surely made this manuscript better quality. Listed below states the modifications we have made or our explanations to some of the reviewers’ comments:
[Note: For easy reference, the reviewers’ comments are cited below in black, whereas our replies are in blue]
Thank you for your pertinent comment and suggestion. A revised version of our manuscript has addressed all comments. A separate document explaining the changes made in the revised manuscript and how these changes effectively addressed the reviewers’ comments has also been provided. We would like to thank you for spending time and effort on this reviewing process, which has surely improved the quality of this manuscript.
- “At the end of the Abstract, what is the meaning of the sentence “Approximately 80% of the data compression can be achieved in physiological signal acquisition applications”?”
Thanks for this question.
The words "data compression" in the abstract are difficult to understand. Therefore, the last sentence in the abstract has been modified. The Compression Ratio (CR) in Ref. [26] has been used in the revised version. The details are presented in Section 4. As shown in Ref. [26], CR is the ratio of the average number of bits/sec in the signal obtained from a uniform ADC to the average number of bits/sec in the signal obtained from an LC ADC. Compared to a conventional uniform sampling ADC, the proposed level-crossing sampling ADC reduces power consumption by over 80% when acquiring physiological signals such as ECG. The value is calculated as [power(uniform sampling)-power(level-crossing sampling)]/power(uniform sampling), indicating that level-crossing sampling results in power savings. The amount of data captured per second by the LC-SAR hybrid ADC cannot be explicitly determined for a random ECG signal, but can be estimated by observing the power consumption over a period of time. The conventional uniform sampling method requires 3.92 μW to acquire a 900-ms ECG signal, whereas the proposed level-crossing sampling ADC consumes only 0.49 μW. The average rate of data acquisition per second was estimated based on the power consumption. Thus, the CR was approximately eight times.
- “In Introduction, there is no reference from line 32 of page 1 to line 68 of page 2. Why?”
Thank you for your comment. We did indeed miss the citation of references in this section. A revised version of the manuscript has been added. This can be found in the revised version of the manuscript.
- “The word styles of “Figure 1a-b” and “Figure 1b” do not match the template of this Journal. Please correct them with Fig. 1(a) or Fig. 1(b)”
Thanks for this suggestion.
To comply with this suggestion, we have made corrections in the newly revised manuscript. Thank you for pointing out this oversight
- “In Eq. (3), what is FSLCADC?”
Thanks for this inspiring question.
FSLCADC represents the sampling frequency of the LC ADC. The manuscript lacks a sufficient explanation, which hinders the reader's understanding. This has been revised in Section 2.1. Thank you very much for reminding us of this issue.
- “In Eq. (9), why do you use (3*epsilon)?”
Thanks for this inspiring question.
This assertion is grounded in the three-sigma rule for the normal distribution, which states that "virtually all" values fall within a range of three standard deviations, plus or minus the mean. In other words, conducting experiments within this range would be feasible. Thus, it is possible to regard a 99.7% chance as "almost certain" in an experimental setting. By determining the unit capacitance size in accordance with this regulation, the mismatch is minimized to the greatest extent feasible so as not to compromise the overall accuracy of the converter.
A detailed analysis has been added in the new version, and it is assumed that the unit capacitance C follows a normal distribution with a mean value C0 and standard deviation σc. Therefore, for m capacitors connected in parallel, the total capacitance size is mC0 + m1/2σc. For an N-bit converter, to reduce the error caused by the capacitance mismatch to less than LSB/2, the following equation should be satisfied:
|
(9) |
(9) |
For an N bits segmented capacitor array with M bits high segmentation, Cmsb and Ctotal are, respectively,
|
(10) |
(10) |
|
(11) |
(11) |
Therefore, by combining Eq. (9), we can determine that the unit capacitance size of the segmented capacitor array should satisfy Eq. (12).
|
(12) |
(12) |
When the standard deviation of the capacitance of the highest bit is positive and that of the rest of the capacitance is negative, Eq. (13) can be derived from the worst-case scenario.
|
(13) |
(13) |
The standard deviation of the capacitance followed a normal distribution. To enhance the reliability of the converter in this study, the three-sigma rule of the normal distribution was utilized to determine the unit capacitance size C. By using the equation ε=σc/C0 and combining it with Eq. (13) and the three-sigma rule, it can be determined that the size of the segmented capacitance must meet the following equation
|
(14) |
(14) |
In the revised manuscript, original Eq. (9) is ordered as Eq. (14). An error was found, and Eq. (14) ( Eq. (9)) should be:
This has been updated in the revised version of the manuscript. Thank you for your inquiries, and this paper will now be more credible and effective.
- “In the comparison of performance, why does the largest chip area of 1.167 mm2?”
Thanks for this question.
There are two main reasons for this finding. First, from Table 2, one can be seen that although the proposed LC-SAR ADC has a slightly larger area, its accuracy is 6+11, surpassing other LC ADCs. Second, owing to the needs of the project, the proposed converter uses a 180 nm process, which will have a relatively larger area compared to ADCs using the 130-40 nm process in other studies. Obviously, if more advanced processes are used, the proposed ADC will have a smaller chip area.
It is worth noting that the proposed ADC has better FoMs.
- “The performance comparison shows that this study is novel. However, the authors need to provide the measured results. I need to make sure that the proposed LC-SAR works correctly”
Thanks for the pertinent remarks and suggestions.
To address this comment, Monte Carlo simulation results for the comparator offset voltage and for the overall converter at various process angles were added. Fig. 9, Fig. 14, and Fig. 20 in the revised manuscript are the corresponding supplements. Meanwhile, owing to the budgetary limitations of this project, the proposed design is not yet fully developed. Once we obtain a supportive fund, we proceed with the proposed design for silicon. Nevertheless, we still hope to provide valuable results and justifications by performing careful post-layout simulations under all types of conditions and process corners.
Finally, we would like to thank you for spending time and efforts on this reviewing process.
Sincerely yours,
Weilin Xu, Ph.D.
Professor in Key Laboratory of Microelectronic Devices and Integrated Circuits,
Guilin University of Electronic Technology

Reviewer 2 Report
Comments and Suggestions for Authors
The paper describes a hybrid ADC converter. The concept presented is not new. Such converters were described in the literature more than 10 years ago. The authors have insufficiently explained what the innovation of the proposed solution is based on. In my opinion, the innovation lies in the use of a SAR converter instead of Flash, which was used in previously known solutions. The paper lacks a solid analysis of the impact of inaccuracy of geometric sizes and technological parameters on the accuracy of the converter. In the paper, the analysis was based on formula (9), which was neither derived nor cited from the literature. For this reason, it is difficult to assess the reliability of such an analysis. In the presented simulation results, no Monte Carlo analysis results are given that confirm that the assumed processing accuracy can be achieved at all in this converter. Moreover, the converter's processing errors not only depend on the accuracy of the capacitor array. Accuracy is also strongly influenced by the offset voltage of the comparators. Neither the schematics of the comparators nor the parameters of the transistors used in the comparators are shown, for this reason it is difficult to judge what offset voltage they can have. The comparison of the parameters of the converters given in Table 2 is not fair because: a) it is not given which converter is just a design and which is a prototype fabricated in silicon; b) the proposed converter is powered by a much higher voltage (1.8V) than the others, which limits its application in IoT devices
Author Response
Mar. 5, 2024
Dear reviewer 2,
Please find submitted the revised version of manuscript “electronics-2882271”, entitled “A 0.49–4.34-μW LC-SAR Hybrid ADC With 10.85-bit ENOB and 20 KS/s
Bandwidth”, authored by Hai Tang, Haiou Li, Baolin Wei, Xueming Wei and me.
We have incorporated the Editor’s and referees’ recommendation into the revised manuscript. We highly appreciate the valuable time and pertinent remarks and suggestions from the reviewers, which have surely made this manuscript better quality. Listed below states the modifications we have made or our explanations to some of the reviewers’ comments:
[Note: For easy reference, the reviewers’ comments are cited below in black, whereas our replies are in blue]
Thank you for your pertinent comment and suggestion. A revised version of our manuscript has addressed all comments. A separate document explaining the changes made in the revised manuscript and how these changes effectively addressed the reviewers’ comments has also been provided. We would like to thank you for spending time and effort on this reviewing process, which has surely improved the quality of this manuscript.
- “The paper describes a hybrid ADC converter. The concept presented is not new. Such converters were described in the literature more than 10 years ago. The authors have insufficiently explained what the innovation of the proposed solution is based on. In my opinion, the innovation lies in the use of a SAR converter instead of Flash, which was used in previously known solutions.”
Thanks for this inspiring comment.
The novelty of the proposed solution is not sufficiently explained, especially in the background description and literature review in Section 1. To address this issue, we have incorporated the necessary information into the revised manuscript.
In fact, the literature [21] you mentioned that the flash-SAR hybrid ADC architecture has indeed inspired us.
The sampling clock of the hybrid architecture SAR ADC proposed in this study was generated by the LC ADC. Additionally, the SAR ADC is required to transmit the output result to the LC ADC to assist it in completing level-crossing detection. This effectively reduces the power consumption of the LC ADC by circumventing the drawbacks of the conventional LC ADC, which has a very high operating frequency and high-power consumption during the conversion process. Based on the findings presented in Section 4, the proposed LC-SAR hybrid architecture ADC operates at a power consumption merely 0.36 μW, significantly lower than alternative configurations, even when an input signal of 14.975 kHz is employed.
The main purpose of the hybrid ADC architecture proposed in this study is to accommodate the low-power consumption requirements of IoT and biomedical detection applications. Despite not requiring an external clock, a conventional LC ADC must operate at a very high frequency to meet the instantaneous rate of change of the input signal and ensure accuracy. Furthermore, the charge scaling circuit in the LC ADC typically comprises capacitors, resulting in large dynamic power consumption. Furthermore, the architecture described in Ref. [21] still has this limitation.
Therefore, in this study, an external low-frequency clock was used to replace the high-frequency clock of a traditional LC ADC. To meet the requirement of an instantaneous rate of change for the external input signal, an additional ADC is implemented, capitalizing on the property that the output of the previous conversion remains valid prior to the subsequent conversion. Based on the application scenario described in this paper, the most suitable additional ADC is the SAR ADC, which is characterized by its low power consumption and high accuracy. An LC ADC requires the output of this additional ADC to complete the level-crossing detection process during operation.
The level-crossing procedure in this LC ADC is distinct from the conventional LC architecture. As illustrated in Fig. 6 of Section 3.1, the LC ADC lacks an up/down counter. Therefore, to ascertain whether a level-crossing transpired, it is necessary to compare each conversion to the preceding conversion. In other words, the designed LC ADC detects level crossings by calculating the difference between the inputs at the approximate previous moment (t-1) and the current moment (t).
Some content has been added to the revised manuscript to discuss its innovation so that readers can better understand it. Thank you for your inquiries, and the revision has surely improved the quality of this manuscript.
- “The paper lacks a solid analysis of the impact of inaccuracy of geometric sizes and technological parameters on the accuracy of the converter. In the paper, the analysis was based on formula (9), which was neither derived nor cited from the literature. For this reason, it is difficult to assess the reliability of such an analysis.”
Thanks for the pertinent remarks and suggestions.
Indeed, this study failed to provide a comprehensive analysis of the variables that influence the precision of a converter and neglected to describe the inception of Formula 9. To adhere to this recommendation and enhance the credibility of this manuscript while facilitating the reader's comprehension of the designed converter, we have appended the following derivation of Formula 14 (previously Formula 9) to the revised manuscript:
Here, it is assumed that the unit capacitance C follows a normal distribution with mean value C0 and standard deviation σc. Therefore, for m capacitors connected in parallel, the total capacitance size is mC0 + m1/2σc. For an N-bit converter, to reduce the error caused by the capacitance mismatch to less than LSB/2, the following equation should be satisfied:
|
(9) |
(9) |
For an N bits segmented capacitor array with M bits high segmentation, Cmsb and Ctotal are, respectively,
|
(10) |
(10) |
|
(11) |
(11) |
Therefore, by combining Eq. (9), we can determine that the unit capacitance size of the segmented capacitor array should satisfy Eq. (12).
|
(12) |
(12) |
When the standard deviation of the capacitance of the highest bit is positive and that of the rest of the capacitance is negative, Eq. (13) can be derived from the worst-case scenario.
|
(13) |
(13) |
The standard deviation of the capacitance followed a normal distribution. To enhance the reliability of the converter in this study, the three-sigma rule of the normal distribution was utilized to determine the unit capacitance size C. By using the equation ε=σc/C0 and combining it with Eq. (13) and the three-sigma rule, it can be determined that the size of the segmented capacitance must meet the following equation
|
(14) |
(14) |
In the revised manuscript, original Eq. (9) is ordered as Eq. (14). An error was found, and Eq. (14) ( Eq. (9)) should be:
You can see them in Section 3, and these new additions will make the proposed circuit more reliable and help the reader understand them more easily.
- “In the presented simulation results, no Monte Carlo analysis results are given that confirm that the assumed processing accuracy can be achieved at all in this converter. Moreover, the converter's processing errors not only depend on the accuracy of the capacitor array. Accuracy is also strongly influenced by the offset voltage of the comparators. Neither the schematics of the comparators nor the parameters of the transistors used in the comparators are shown, for this reason it is difficult to judge what offset voltage they can have.”
Thanks for this inspiring comment.
Monte Carlo analysis is necessary. In addition to the accuracy analysis of the capacitor array, it was necessary to provide an offset voltage analysis of the comparators. Note that the accuracies of the LC ADC and SAR ADC in the manuscript are different. Therefore, the requirements for the offset voltage performance of their comparators are also different, and we have augmented the recently revised manuscript with Monte Carlo simulation data pertaining to the two aforementioned comparators. These results are shown in Fig. 9 and Fig. 14, respectively. Because of the financial limitations imposed on this undertaking, a silicon stage has not yet been implemented for the proposed design. We will immediately begin the silicon design upon receipt of support funding. Nevertheless , we still hope to provide valuable results and justifications by performing careful post-layout simulations under all types of conditions and process corners. Thus, in the revised manuscript, we have supplemented the overall converter back-end process corner simulation data in Fig. 20. It is expected that the inclusion of supplementary information will improve the credibility of the manuscript and facilitate the reader's comprehension of the designed converter's performance.
- “The comparison of the parameters of the converters given in Table 2 is not fair because: a) it is not given which converter is just a design and which is a prototype fabricated in silicon; b) the proposed converter is powered by a much higher voltage (1.8V) than the others, which limits its application in IoT devices.”
Thanks for this suggestion.
The lack of clear indication on whether these ADCs are the prototypes fabricated in silicon has indeed led to unfair comparisons. Thus, modifications have been implemented in Table 2 of the revised manuscript.
The proposed ADC is a building module in our project. Therefore, it uses a unified 1.8 V power supply. However, one must admit that a higher power supply limits its application in IOT devices. This results in a larger chip area and higher power consumption. The trend for mobile IoT chips is to adopt more advanced processes and lower power supply voltages. Nevertheless, competitive ultralow power consumption, excellent ENOB, and FoMs are still achieved at a supply voltage of 1.8V, owing to the innovation of this study.
Finally, we would like to thank you for spending time and efforts on this reviewing process.
Sincerely yours,
Weilin Xu, Ph.D.
Professor in Key Laboratory of Microelectronic Devices and Integrated Circuits,
Guilin University of Electronic Technology

Round 2
Reviewer 1 Report
Comments and Suggestions for Authors
The authors have revised the resubmitted manuscript according to the reviewers’ comments point by point, and many improvements have been completed with detailed answers. The revised version is good. However, an issue needs to be added, if possible. That is, the authors need to provide the measured results to ensure the proposed LC-SAR system works correctly.
Author Response
Mar. 12, 2024
Dear Reviewer 1,
Please find submitted the revised version of manuscript “electronics-2882271”, entitled “A 0.49–4.34-μW LC-SAR Hybrid ADC With 10.85-bit ENOB and 20 KS/s
Bandwidth”, authored by Hai Tang, Haiou Li, Baolin Wei, Xueming Wei and me.
We have incorporated the Editor’s and referees’ recommendation into the revised manuscript. We highly appreciate the valuable time and pertinent remarks and suggestions from the reviewers, which have surely made this manuscript better quality. Listed below states the modifications we have made or our explanations to some of the reviewers’ comments:
[Note: For easy reference, the reviewers’ comments are cited below in black, whereas our replies are in blue]
Thank you for your pertinent comment and suggestion. A revised version of our manuscript has addressed all comments. A separate document explaining the changes made in the revised manuscript and how these changes effectively addressed the reviewers’ comments has also been provided. We would like to thank you for spending time and effort on this reviewing process, which has surely improved the quality of this manuscript.
- “That is, the authors need to provide the measured results to ensure the proposed LC-SAR system works correctly”
Thanks for this suggestion.
Because of the budget constraints of this project, the designed converter has not yet been implemented in silicon. Thus, the current results are the back-end simulation results. The Monte Carlo simulations have been added to the revised manuscript. We will immediately begin silicon implementation upon receipt of sufficient support funding.
Nonetheless, we hope to provide valuable results and justifications by performing careful post-layout simulations under all types of conditions and process corners.
Based on your comment, Fig. 21 and 22 have been added to the revised manuscript. Fig.21 shows the transient simulation results of the system when the input signal was a sinusoidal signal of 200 Hz. This illustrates that the LC ADC does not detect level crossing when the rate of change of the input signal decreases or when the input signal goes from a high frequency to a low frequency, which causes the SAR ADC to stop sampling and reduces the operating frequency of the LC ADC. Thus, a reduction in system power consumption was achieved through this approach.
Fig. 22 shows the simulation results of the output signal of the LC ADC, which demonstrates the level-crossing detection process. When the CHANGE signal is higher than VH or lower than VL, it indicates that level crossing has been detected. If no level crossing is detected after a certain number of clock cycles, then the operating frequency can be lowered to achieve an adaptive operating clock.
We hope that the supplementary materials in the revised manuscript provide more detailed explanations and ensure that the proposed LC-SAR system works correctly.
Finally, we would like to thank both you and the reviewers for spending time and efforts on this reviewing process.
Sincerely yours,
Weilin Xu, Ph.D.
Professor in Key Laboratory of Microelectronic Devices and Integrated Circuits,
Guilin University of Electronic Technology

Reviewer 2 Report
Comments and Suggestions for Authors
The improvements made are satisfactory.
Author Response
Thank you for spending time and effort on this reviewing process, which has surely improved the quality of this manuscript.